# Microorganisms for Ginsenosides Biosynthesis: Recent Progress, Challenges, and Perspectives

**DOI:** 10.3390/molecules28031437

**Published:** 2023-02-02

**Authors:** Luan Luong Chu, Nguyen Quang Huy, Nguyen Huu Tung

**Affiliations:** 1Faculty of Biotechnology, Chemistry and Environmental Engineering, Phenikaa University, Hanoi 12116, Vietnam; 2Bioresource Research Center, Phenikaa University, Hanoi 12116, Vietnam; 3Faculty of Biology, University of Science, Vietnam National University, Hanoi (VNU), 334 Nguyen Trai, Thanh Xuan, Hanoi 10000, Vietnam; 4National Key Laboratory of Enzyme and Protein Technology, University of Science, Vietnam National University, Hanoi (VNU), 334 Nguyen Trai, Thanh Xuan, Hanoi 10000, Vietnam; 5Faculty of Pharmacy, Phenikaa University, Hanoi 12116, Vietnam

**Keywords:** ginsenosides, *Panax* species, endophytes, engineered microorganisms

## Abstract

Ginsenosides are major bioactive compounds present in the *Panax* species. Ginsenosides exhibit various pharmaceutical properties, including anticancer, anti-inflammatory, antimetastatic, hypertension, and neurodegenerative disorder activities. Although several commercial products have been presented on the market, most of the current chemical processes have an unfriendly environment and a high cost of downstream processing. Compared to plant extraction, microbial production exhibits high efficiency, high selectivity, and saves time for the manufacturing of industrial products. To reach the full potential of the pharmaceutical resource of ginsenoside, a suitable microorganism has been developed as a novel approach. In this review, cell biological mechanisms in anticancer activities and the present state of research on the production of ginsenosides are summarized. Microbial hosts, including native endophytes and engineered microbes, have been used as novel and promising approaches. Furthermore, the present challenges and perspectives of using microbial hosts to produce ginsenosides have been discussed.

## 1. Introduction

Natural bioactive compounds and traditional medicines have great importance in the treatment of prevalent human diseases and health care. Bioactive compounds mainly include various valuable metabolites, such as stilbenes, flavonoids, terpenoids, polyketides, and alkaloids [1,2]. These molecules show multifunctional properties, such as anticancer, antioxidant, antimicrobial, antifungal, and antiviral activities. They also exhibit many beneficial effects on the treatment of neurological, cardiovascular and metabolic, immunological, inflammatory, and related diseases [3]. According to the World Health Organization, approximately 19 million people were newly diagnosed with cancer in 2020, with a total of 10 million cancer-related deaths. the top five cancer diseases, such as lung cancer, female breast cancer, colorectal cancer, stomach cancer, and liver cancer, are speeding up to rank among the top dangerous reasons for the death rate of humankind. Therefore, the investigation and development of approaches for producing a novel anticancer agent have been highly required in human life. Among a lot of natural bioactive compounds, ginsenosides from *Panax* species have been reported to have a wide variety of pharmacological and biological activities, including anticancer activity. Approximately 200 ginsenosides have been identified from the leaves, roots, flower buds, roots, and berries of *Panax* species in recent years [4]. Although ginsenoside has multifunctional properties, it has been considerably attended to the regulation of cancer cell metabolism in vitro and in vivo [5].

The most powerful tool for treating cancer is chemotherapy. Chemotherapy is widely used to treat cancer at all stages of progression in the various cancer cell lines [6]. the extraction from plants and chemical synthesis are two traditional approaches to produce anticancer drugs. These methods are limited because of their disadvantages. While the former method is time-consuming and based on seasonable weather, the latter approach often requires unexpected purity, high reaction pressure, and high temperatures with the use of the harmful reagent [7]. Moreover, the immune response is activated by using chemically synthesized anticancer drugs. the immune response has a negative effect on healthy patients [8]. Thus, simple, safe, and efficient approaches are strongly required for the production of anticancer drugs. Enzymatic reactions are able to produce the specific reaction and a simple alternative; however, it is too expensive. Although the generation of the cofactor has been developed through single-vessel recycling reactions, enzymatic reactions are not easy to produce on a large-scale [9,10]. Moreover, endophytes, which colonized inside plant tissues without displaying signs of disease symptoms, showed the ability to produce enzymes for natural compound biosynthesis, resulting in the method being recognized as safe with a qualified presumption of safety [11]. While endophytic bacteria and fungi are also concerned with a promising novel resource product, engineered microorganisms are used as cell factories for the production of bioactive compounds. Whole-cell microorganisms not only show the ability to catalyze reaction efficiency, but also exhibit a low consideration for environmental problems. Whole-cell biocatalysts are becoming the most preferred approach for the industrial production of anticancer medicine [12]. Moreover, microbial production provides eco-friendly and efficient approaches for the development of anticancer drugs [13,14]. In this review, we describe ginsenoside as a widely used anticancer compound. Furthermore, the application of bacterial and fungal endophytes for the biosynthesis of ginsenoside is summarized. In addition, the engineering of microbes for ginsenoside production is discussed. This review will present several examples of ginsenoside compounds that have been produced from bacterial and fungal endophytes as well as engineered microbes. Finally, the challenges and perspectives of microorganisms for ginsenoside production in sustainable development are discussed.

## 2. Ginsenosides: Classification and Cell Biological Mechanism in Anticancer Activities

Ginsenosides, a triterpenoid glycoside, are major constituents extracted from *Panax ginseng* (*Pg*), *P. notoginseng*, *P. quinquifolium*, and another species belonging to the *Panax* genus, *P. vietnamensis* [15,16]. In spite of the fact that all tissues of ginseng contain ginsenosides, the root is known as a major source of accumulating total ginsenosides. It was demonstrated that there are a lot of factors that impact on the total ginsenosides in the *Panax* species, including the age of the ginseng, plant growth promotion, plant–pathogen interaction, and collection and extraction methods. According to their chemical structure, ginsenosides are divided into four groups: (1) protopanaxadiol (PPD) group consisting of Ra1, Rb1, Rc, Rd, R1, and compound K (CK). the sugar moiety attaches at *β*-OH of C-3 and/or C-20 in PPD compounds; (2) protopanaxatriol (PPT), consisting of Re, Rf, Rg1, and Rh1. PPT has a sugar moiety that binds to α-OH of C-6 and/or *β*-OH of C-20; (3) the ocotillol group, including majonoside R2, vinaginsenoside R1, and pseudoginsenoside F11. This group processes a five-membered epoxy ring at C-20; (4) oleanane ginsenoside group consisting of a pentacyclic triterpene skeleton, such as Ro and R_OA_ (Table 1) [17,18]. On the other hand, ginsenosides are also classified into two groups based on the percentages of the total ginsenoside content in each *Panax* species. While a major ginsenoside has been determined to account for over 80% of total ginsenoside content, minor ginsenosides have been present in low concentrations in wild and red ginseng (Figure 1). the major ginsenosides, such as Rb1, Rb2, Rc, Rd, Re, Rg1, and R1, can be digested into minor ginsenosides by hydrolyzing the multiple sugar moieties. Although the presence of sugar moieties increases the stability and solubility properties of compounds, these residues lead to a decrease in the permeability of the cell membrane. Therefore, the minor ginsenosides, such as F1, F2, Rg2, Rg, Rh2, and CK, have higher anticancer activities than the major ginsenosides [19]. Ginsenoside Rg3 and Rh2, which possess two and one sugar moieties, respectively, show potent apoptotic and antiproliferative activities [20]. However, ginsenosides, which possess three or more sugar residues, including Rd, Rc, and Rb1, show little or no sign of antiproliferative activity [21]. Moreover, many minor ginsenosides show great biological and pharmacological activities. the novel minor 5,6-didehydroginsenoside Rg3 from Pg showed a significant effect on anti-inflammatory activity [22]. CK has been shown to have antidiabetic, antitumor, anti-inflammatory, and hepatoprotective activities [23]. A new derivative of ginsenoside from *P. notoginseng*, 25-OCH_3_-PPD, not only inhibited various types of cancer cell lines, such as pancreatic, breast, and lung cancer, but also exhibited nongenotoxic properties for antitumor treatment [24]. Noticeably, differences in the position of sugar linkers and hydroxyl groups give direction to biological activities. It has been demonstrated that the available sugar moieties at C-6 of ginsenosides have less anticancer activity than linkages at C-3 or C-20 of ginsenosides. the moieties at C-6 of ginsenosides decrease the interaction between these compounds and their binding proteins by blocking the way into the binding pocket. For example, ginsenoside Rh2 exhibits stronger potency in anticancer activity than Rh1 [25]. In addition, the interaction between ginsenoside and *β*-OH of the cholesterol in the cell membrane is affected by the number and size of the hydroxyl group, and due to polar compounds, this could change membrane fluidity and function [26].

Ginsenosides exhibited various pharmacological activities, such as antimicrobial, antiaggregant, antioxidant, prevention of cardiovascular disease, and improving immune function [30]. Ginsenoside has been used as a source for the cosmetic and food industries. However, it is the most famous compound as an anticancer agent. While major ginsenosides have no significant or only weak antiproliferative and antiangiogenic activities, minor ginsenosides exhibit strong anticancer activities and cause cell death [19,21]. Minor ginsenosides have a significant inhibitory effect on cell proliferation and differentiation. the possible cell biological mechanisms of action of ginsenosides based on the existing research are listed as follows: effect on proliferation and differentiation, regulation of cell cycle and p53/p21/murine double minute-2 pathways, modulation of cell death (Bcl-2, Bcl-xL, inhibitory apoptotic protein (IAP), caspases, and death receptors), modulation of growth factor and protein kinase, inflammatory response molecules, and effects on DNA damage (Figure 1) [19,21]. As an example, Rh2 has been shown to inhibit the proliferation of MCF-7 human breast cancer cells via inducing a G1 arrest in cell cycle progression, which is associated with the enzyme expression of the cyclin-dependent kinase (CDK) inhibitor protein p21 [27]. Rh2 also inhibits cell growth and induces apoptosis of human leukemia (HL-60) cells via the tumor necrosis factor-α (TNF-α) pathway. Similarly, Rh2 and Rg3 repress cell proliferation and induce apoptosis in HL-Jurkat cells by increasing a mitochondrial reactive oxygen species (ROS) [25]. Moreover, Rg5 inhibits cell proliferation in retinoblastoma cells by downregulating BCL2 expression in the protein kinase B (PI3K/AKT) signaling pathway [31]. In addition, Rh1 was the most effective at causing differentiation of F9 teratocarcinoma stem cells by binding to a glucocorticoid receptor. Rh1 also exhibits the ability to inhibit the proliferation, migration, and invasion of colorectal cancer cell lines. the inhibition was achieved by the repression of matrix metalloproteinase 1 (MMP1) and expression of metalloproteinase 3 (MMP3) expression level [28]. Ginsenoside F1 enhances natural killer (NK) cell cytotoxicity in cancer immunosurveillance by insulin-like growth factor-1 (IGF-1) treatment [29]. Noticeably, CK not only increases the mRNA levels of angiogenic inhibitors thrombospondin-1 (TSP-1), -1 MMP-1, and metalloproteinase-2 (MMP2), but also decreases the mRNA levels of angiogenic factors vascular endothelial growth factor A (VEGF-A) and fibroblast growth factor 2 (FGF-2) in 3T3-L1 adipocytes. Furthermore, CK shows the ability to inhibit and migrate human glioblastoma U87MG and U373MG cells via an arrested cell cycle progression at the G0/G1 phase [23]. Moreover, 25-OCH_3_-PPD has significant effects on decreasing the survival and inhibiting the proliferation of human prostate and breast cancer cell lines [24]. Thus, ginsenosides likely have therapeutic potential for the treatment of various cancer diseases.

## 3. Endophytes as Novel Biological Source of Ginsenosides

Endophytic microbes spend all or part of their life cycle in plant tissues without causing any symptoms of the disease [11]. Endophytes are distributed in the roots, stem, petiole, leaves, peduncle, buds, seed, berry, inflorescences of weeds, and fruit. Endophytes include fungi, bacteria, and actinomycetes, which form a symbiotic relationship with their plant hosts. It has been demonstrated that endophytes are excreted metabolites to attack foreign pathogens or lysis cells. Endophytes are used as biological agents for inducing and promoting growth. the protection mechanisms in plant life, such as the biosynthesis of phytohormones, the production of siderophores, nitrogen fixation, and phosphorus, are induced by endophytes. Moreover, endophytes also exhibited the ability to degrade xenobiotics via enzymatic activities [32,33]. the characteristics of endophytes showed the ability to produce the same or similar natural compounds as their plant hosts. Natural compounds are synthesized from the various metabolic pathways and classified as glycosides, alkaloids, polyketides, terpenoids, coumarins, stilbenoids, flavonoids, lipids, and phospholipids. These secondary metabolites exhibited the various biological activities of anticancer drugs, antibiotics, antiviral drugs, antioxidants, insecticidal activities, antidiabetic agents, and immune suppressive compounds [34,35]. the possible reasons behind the coproduction of endophytes and their plant hosts are independent of the evolution of the natural biosynthesis pathway. Moreover, it is hypothesized that the simultaneous biosynthesis of natural compounds between the plant host and its endophytes is an effect of horizontal gene transfer, even though this mechanism has only been reported between endophytic microbes [36]. In order to clearly understand the similar secondary metabolites biosynthesis, the relationship between plant hosts and endophytes requires extensive research. Furthermore, it is demonstrated that novel bioactive compounds and metabolic pathways could be achieved by using a different type of endophyte. These metabolites are produced via stereoselective or region-selective reactions [37]. Therefore, endophytes are extremely important in the production of novel bioactive compounds such as ginsenosides.

### 3.1. Bioproduction of Ginsenosides by Native Endophytes

Since ginseng endophytes are concerned with a novel source for producing ginsenosides, the number of isolated endophytes from *Panax* species has increased [38,39]. Pg is one of the most useful resources for isolating the endophytes with the potential of producing ginsenosides (Figure 2A). As a result, 38 endophytic fungi have been isolated from *Pg*, and the capacity of ginsenoside production has been identified. *Nectria*, *Aspergillus*, and *Penicillium* species are the three dominant endophytes from Korean ginseng. Other endophytes, such as *Ascomycete, Cladosporium*, *Engyodontium*, *Fusarium*, *Plectoshaerella*, and *Verticillium* sp., are also presented with the potential of producing ginsenosides. the highest concentration of total ginsenosides was 0.181 mg/mL by *Fusarium* sp., followed by 0.144 mg/mL by *Aspergillus* sp. and *Verticillium* sp. [40]. Similarly, the isolated endophyte from *P. notoginseng* is also concerned with the promising potential of producing ginsenosides (Figure 2A). Among 53 endophytic fungi isolated from the seeds and roots of *P. notoginseng*, two novel strains, *Fusarium* sp. PN8 and *Aspergillus* sp. PN17, exhibited to produce 1.061 and 0.583 mg/mL of total ginsenosides, respectively. the analysis data identified that *Fusarium* sp. PN8 produced Rb1, Rd, and Rg3, and *Aspergillus* sp. PN17 synthesized Re, Rd, and Rg3 [39]. Interestingly, the first ginsenoside-producing endophytic fungi were isolated from *Aralia elata* grown in Northeast China (Figure 2A) [41]. *Aralia elata* belongs to the family *Araliaceae*, which could produce ginsenosides. the various endophytic fungi from *Aralia elata* have been found to produce a small number of ginsenosides, including *Alternaria*, *Diaporthe, Penicillium, Fusarium*, *Trichoderma*, *Botryosphaeria*, *Camarosporium*, *Cryptosporiopsis*, *Dictyochaeta*, *Nectria*, *Peniophora*, *Schizophyllum*, and *Cladosporium.* Among them, the highest concentration of ginsenoside (Rb2 and Re) was only produced at 2.049 mg/mL by *Penicillium* spp [41]. On the other hand, endophytic bacteria have been reported to produce ginsenoside. Among 81 bacteria isolated from *Pg*, three endophytic bacteria showed the capacity to produce rare ginsenosides such as F2, Rh2, and Rg3. the highest concentrations of Rg3 and Rh2 were achieved by the genus *Agrobacterium* with 62.20 mg/L and 18.6 mg/L, respectively. Noticeably, the genus *Agrobacterium* shared many characteristics with *Agrobacterium rhizogenes.* This is the first time that *Pg* endophytic bacteria have been shown to produce ginsenosides Rg3 and Rh2 [42].

### 3.2. Biotransformation of Major to Minor Ginsenosides by Endophytes

Unfortunately, ginsenoside production by endophytes is still extremely low, and it seems not to be enough for amplification in industry. Therefore, the production of ginsenosides via biotransformation is a suitable alternative to obtain more valuable compounds. Endophytic microorganisms are used in biotransformation to convert the major to rare ginsenosides. the diversity of ginsenoside structures explains the variation in the number, type, and position of sugar residues the sugar component could comprise various types of sugars, such as *β*-D-glucose, *β*-D-xylose, *α*-L-rhamnose, *α*-L-arabinose (pyranose), *α*-L-arabinose (furanose), and *β*-D-glucuronic acid [43,44]. While sugar residues are attached to the *β*-OH at C-3 and or C-20 in the PPD group, they are attached to the *α*-OH at C-6 and/or *β*-OH at C-20 in the PPT group. In the PPD ginsenosides, *β*-D-glucopyranose could attach to both the C-3 and C-20 positions. *β*-D-glucopyranosyl-(1→2)-*β*-D-glucopyranose is linked to C-3, while other saccharide residues, such as *α*-L-arabinopyranosyl-(1→6)-β-D-glucopyranose and α-L-arabino-furanosyl-(1→6)-*β*-D-glucopyranose, are attached to C-20. In the PPT group, C-20 is the exclusive position for attaching the *β*-D-glucopyranose, whereas mostly *β*-D-glucopyranosyl-(1→2)-*β*-D-glucopyranose *α*-L-arabinopyranosyl-(1→6)-*β*-D-glucopyranose, and *β*-D-xylopyranosyl-(1→2)-*β*-glucopyranose are linked to C-6 [45].

Based on understanding the chemical structure of ginsenosides, the conversion from major to rare ginsenosides could be prepared through deglycosylation and hydrolysis reactions. These reactions are catalyzed by various glycosidases, including *β*-glucosidase, *β*-galactosidase, *β*-xylosidase, cellulase, *α*-L-arabinopyranosidase, and *α*-L-arabinofuranosidase [45]. *β*-glucosidase is known as the best hydrolyzer of glycosidic bonds on the ginsenosides among the various glycosidases that demonstrated the ability to catalyze deglycosylation and hydrolysis reactions. the first biotransformation of major ginsenosides using an endophyte isolated from *P. notoginseng* was reported in 2013 (Table 2). In this study, among 136 endophytes, *Fusarium oxysporum* or *Fusarium* sp. could produce glucosidase to hydrolyze Rb1 to F2 and CK. Moreover, *Nodulisporium* sp. could convert Rh1 to Pseudo-ginsenoside RT4, while *F. oxysporum* catalyze Rh1 to PPT [38]. In another case, the conversion from Rb1 to Rd was accomplished by *Trichoderma koningii* and *Penicillium chermesinum* at a rate of 40.00 and 74.24% under shake-flask culture, respectively. Both endophytes were isolated from *P. notoginseng* [46]. Another example is the isolation of 15 *β*-glucosidase-producing endophytic fungi from *Pg* roots (Table 2; Figure 2B). the highest *β*-glucosidase activity for converting major ginsenoside Rb1 to minor ginsenoside CK compound belongs to strain GE 17-18, *Arthrinium* sp. the metabolic pathway for CK biosynthesis was shown as Rb1 → Rd → F2 → CK: in the first stage, Rb1 is converted to Rd through a deglycosylation reaction from gentiobiose (O-*β*-glucopyranosyl(1→6)-*β*-D-glucopyranose). After that, Rd cleaves at (O-*β*-glucopyranosyl (1→2)-*β*-D-glucopyranose to form F2. In the final step, the formation of CK from F2 is achieved through hydrolyzing *β*-D-glucopyranose attached to the C-3 position [47].

On the other hand, the first β-glucosidase-producing endophytic bacteria were reported in 2017. Strain GE 17-7 was isolated and identified as *Burkholderia* sp. for the biotransformation of Rb1 to Rg3 via Rd (Rb1 → Rd → Rg3) (Table 2; Figure 2B). In the first stage, *β*-(1-6)-glucosidase-producing *Burkholderia* sp. is catalyzed to form the Rd by attacking the outer *β*-(1-6)-glycosidic linkage at the C-20 position without cleaving any *β*-D-glucosidic linkages. Rd is then converted to Rg3 through a specific hydroxylation of the glycosidic linkage at the C-20 position [48]. Similarly, the strain GE 32 was isolated and identified as *Flavobacterium* sp. for producing Rg3 and Gyp-XVII from Rb1. the metabolic pathways for Gyp-XVII and Rg3 biosynthesis were identified as Rb1 → Gyp-XVII and Rb1 → Rd → Rg3, respectively (Figure 2B). Glucosidase-producing *Flavobacterium* sp. not only exhibited the capacity to attach to the C-20 position of Rb1 and then Rd to form Rg3, but also showed the ability to hydrolyze the glucose from gentiobiose (O-*β*-D-glucopyranosyl-(1→6)-*β*-glucopyranose) at the C-3 position of Rb1 to form Gyp-XVII [49].

Interestingly, an endophyte JG09, which was isolated from *Platycodon grandiflorum*, was capable of converting major to minor ginsenosides. Among 69 endophytes from *P. grandiflorum*, 32 endophytes could produce *β*-glucosidase, which catalyze the formation of minor ginsenoside from major ginsenoside. the highest efficiency of conversion from ginseng total ginsenoside and ginsenoside monomers (Rb1, Rb2, Rc, Rd, and Rg1) to minor ginsenosides (F2, CK, and Rh1) was catalyzed by *Luteibacter* sp. and identified as Rb1 → Rd → F2 → CK (Figure 2B). In this pathway, the initial stage is to hydrolyze the β-(1-6)-glycosidic linkage at the C-20 position of Rb1 to synthesize Rd, and the latter stage produces F2 through hydrolyzing-(1-2)-glycosidic linkage at the C-3 position of Rd. the CK is formed by deglycosylation of F2 at the C-3 position. the second and third biotransformation pathways to form CK were provided as Rb2 → CO → CY → CK and Rc →CMc1 → CMe → CK, respectively. In these pathways, the formation of CY and CMc was performed by deglycosylation at the C-3 position of Rb2 and Rc, respectively. Then, the hydrolyzing of the *ββ*-(1-6)-glycosidic linkage at the C-20 position of CY and CMe resulted in the production of CK. Moreover, *β*-glucosidase-producing *Luteibacter* sp. also exhibited the ability to convert Rg1 into Rh1 through deglycosylation at the C-20 position [50]. Importantly, it also demonstrates that *β*-glucosidase-producing *Luteibacter* sp. can develop a specific bioconversion process to obtain minor ginsenosides. Noticeably, endophytes *F. oxysporum*, *Nodulisporium* sp., or *Bacillus* sp. isolated from *P. notoginseng* could produce oxidase to oxidize Rg1 to vinaginsenoside R22. Similarly, *Nodulisporium* sp. could transform Re to vinaginsenoside R13 via oxidative reactions (Table 2) [38]. Recently, the conversion from Rc to Rd was carried out by endophytic *Bacillus* sp. G9y isolated from *Panax quinquefolius* [51]. Interestingly, *Panax bipinnatifidus* var. *bipinnatifidus*, has been reported as a novel fungus with potent *β*-glucosidase for the production of minor ginsenosides [52].

## 4. Ginsenoside Biosynthesis in Engineered Microorganisms

### 4.1. Ginsenoside Biosynthesis in Engineered Bacteria

Engineering microbes have promising alternative strategies for producing ginsenosides from renewable energy sources. Microbial production hosts have been developed as a primary approach for producing ginsenoside on an industrial scale. Small genome size is the best advantage of microorganisms in comparison to plants. Moreover, the metabolite transports between enzymatic chains are not large enough to consider because fewer intracellular organelles are present in microbial cells than in plant cells. Microbial hosts, including bacteria (*Escherichia coli, Corynebacterium glutamicum*, and *Lactococcus Lactis*) and yeasts (*Saccharomyces cerevisiae*, *Yarrowia lipolytica*, and *Pichia pastoris*), provided a promising strategy for producing a high quantity of ginsenosides. While bacteria have been used to produce ginsenoside via recombinant *β*-glucosidase or uridine diphosphate glycosyltransferase (UGTs), yeasts have been metabolically engineered through heterologous gene expression and enzyme engineering [53]. Although *E. coli* lacks the endoplasmic reticulum required for plant membrane-bound cytochrome P450 enzymes (CYP450s) and is less efficient in precursor supplies for the methylerythritol phosphate (MEP) pathway, *E. coli* is known as the most important microbial host for industrial production. *E. coli* has many dominant characteristics compared to plant cells, such as fast growth with high cell density cultivation, controllability under laboratory conditions, well-characterized genetics, and well-developed genetic manipulation technology [54]. In this case, *E. coli* is one of the most common hosts to produce minor ginsenosides via recombinant β-glucosidase expression. the genes coding for β-glucosidase could be isolated from various microorganisms, such as *Microbacterium* sp. Gsoil 167 [55], *Thermotoga thermarum* DSM 5069 [56], *Caldicellulosiruptor bescii* [57], *Niabella ginsenosidivorans* BS26 [58], and *Saccharibacillus kuerlensis* [59]. Recently, the novel and thermostable β-glucosidase has been used to produce high-value minor ginsenosides. For example, β-glucosidase from *Caldicellulosiruptor bescii* was able to completely convert all PPD-type ginsenosides into CK. Interestingly, a chaperone system was used to prevent aggregation and promote refolding of the misfolded protein in expressed-β-glucosidase *E. coli.* Along with the permeabilized cells, chaperone coexpressed-β-glucosidase *E. coli* showed a 2.6-fold increase up to 1.87 mg/mL of CK as compared with nontreated cells [60]. Although biotransformation pathways have been well converted from major to minor ginsenosides in *E. coli*, in recent years, *C. glutamicum* and *L. lactis* have been used to produce ginsenosides. One of the best advantages of *C. glutamicum* and *L. lactis* is that they are generally recognized as safe (GRAS). Both strains are important organisms for the biotechnological industry. While *L. lactis* is known as a traditional fermented microorganism for nutritious food, *C. glutamicum* is used as a producer of amino acids on a million-ton scale [61,62]. It has been reported that *L. lactis* subsp. *cremoris* NZ9000 was used to express β-glucosidase genes from *Paenibacillus mucilaginosus* (*BglPm*) and *Flavobacterium johnsoniae* (*BglBX10*). Production CK from Rb was catalyzed by *BglBX10* with a 70% conversion yield [63]. Similarly, *C. glutamicum* ATCC13032 was used to express *Microbacterium testaceum* ATCC 15829 β-glucosidase. Engineered *C. glutamicum* produced 7.59 g/L of CK and 9.42 g/L of F1 from PPD- and PPT-type ginsenoside mixtures in 24 h, respectively [64]. These results indicated that *C. glutamicum* and *L. lactis* could be used instead of *E. coli* to produce the ginsenoside. It is believed that all alternative approaches to increase the production of ginsenoside used in *E. coli*, including codon optimization and the introduction of a chaperone coexpression system, could be applied to *C. glutamicum* and *L. lactis* for high-production target compounds.

### 4.2. Ginsenoside Production in Engineered Yeasts

The model yeast *S. cerevisiae* and nonconventional yeasts, including *P. pastoris* and *Y. lipolytica*, have been engineered to produce ginsenosides in recent years. All strains are (GRAS) microorganisms for industrial application; however, *S. cerevisiae* is known as the most commonly used yeast [65,66]. Unlike the bacterial model *E. coli*, plant- or mammal-derived enzymes, CYP450s and UGTs, could be easily expressed in *S. cerevisiae*. the reason is that *S. cerevisiae* possesses redox systems that support the approximate environment for functional expression of plant- or mammal-derived enzymes. Moreover, similar to *E. coli*, *S. cerevisiae* is well characterized genetically. Therefore, it eases the development of genetic manipulation technology. As a result, the skeletons of ginsenosides could be modified through glycosylation and hydroxylation reactions [18]. Overexpression of all biosynthesis ginsenoside genes in yeast cells is one of the most general strategies to produce ginsenosides. the key enzymes for ginsenoside biosynthesis from *Pg*, including CYP450s, UGTs, and oxidosqualene cyclases (OSCs), were introduced to achieve ginsenosides in yeasts. Three genes have been found from multiple sources, such as higher plants *Arabidopsis thaliana*, *Sorghum bicolor*, *Solanum lycopersicum*, *Glycine max*, and *Oryza sativa* [67,68]. However, *Pg* has been demonstrated as the most important source for producing ginsenosides with high efficiency. 2,3-Oxidosqualene was converted to dammarenediol (DM) by DM-II synthase (PgDDS) and *β*-amyrin by *β*-amyrinsynthase (*β*-AS). Then, CYP450s were catalyzed to produce the ubiquitous aglycon of ginsenosides, including oleanolic acid (OA), PPD, and PPT. A total of 414 CYP450 genes have been found in *Pg*; however, only 9 genes have been characterized as contributing to ginsenoside biosynthesis. In general, expression of Pg protopanaxadiol synthase (PgPPDS) along with Pg cytochrome P450 reductase (PgCPR1) could convertDM-II to PPD [69]. This synthetic step is one of the most important for the biosynthesis pathway of ginsenosides. Finally, the ginsenoside and its analogs were synthesized by UGTs, which is the final reaction in ginsenoside biosynthesis. Noticeably, the diverse ginsenosides could be generated because of the chemo-, stereo-, regioselectivity, and flexibility activities of UGTs (Figure 3). For instance, UGTPg1 showed the flexibility activities of glycosyltransferase (GTs). While the PPD-producing chassis strain of *S. cerevisiae* harboring UGTPg1 produced 240 µg/L of CK, PPT-producing engineered strains biosynthesized Rh1 with a title of 92.8 mg/L through the glycosylation at the hydroxyl group of C-20 of PPT [70,71]. Furthermore, F2 and Rd can be obtained from Rg3 and Rh2 by expressing UGTPg1 in the PPD-producing chassis strains of *S. cerevisiae*, respectively. UGTPg1 also demonstrated the ability to convert DM to 20S-O-Glc-DM [72]. Interestingly, UGTPg1 overexpressed along with PgUGT74AE2 could produce 3β-20S-Di-O-Glc-DM [73]. In another study, UGTPg45 catalyzed the conversion of PPD to Rh2 by transferring the glucose moiety to the hydroxyl group at C-3. Moreover, Rh3 could be generated from Rh2 by UGTPg29 through the 1-2 glycosidic bond of the hydroxyl group at C-3. PPD-producing chassis strain of *S. cerevisiae* carried out UGTPg45 and UGTPg29 to produce Rh2 and Rh3 with 1.45 and 3.49 µmol g/L DCW, respectively (Table 3) [74]. Recently, a rhamnosyltransferase PgURT94 has been elucidated and showed glycosylation through the transfer of a rhamnose moiety at C6-O-Glc. Therefore, engineered *S. cerevisiae* harboring CYP716A53v2, PgUGT71A54 along with PgURT94 could produce 1.3 g/L of Rg2. While PgUGT71A54 catalyzed the glycosylation by transferring a glucose moiety to the hydroxyl group at C-6, PgUGT71A53 showed the glycosylation at the hydroxyl group at C-20 of PPT. Therefore, the complete biosynthetic pathway of Re was reconstructed in *S. cerevisiae.* Rg2-producing engineered *S. cerevisiae* CYP716A53v2, PgUGT71A53, PgUGT71A54, and PgURT94 could be produced at 3.6 g/L of Re (Figure 3) [75]. Noticeably, novel UGTs have been reported from bacteria in recent years, such as *B. licheniformis* DSM-13, *B. subtilis* CGMCC [76]. A flexible GT from *B. licheniformis* DSM-13 not only uses flavonoids and stilbenes, but also holds potential in the biosynthesis of unnatural ginsenosides. It is demonstrated that YjiC from *B. subtilis* can catalyze the glycosylation on the hydroxyl group at C-3 and then C-12 of PPD to produce Rh2 and F12 with 2.1% and 97.8% of conversion, respectively (Table 3) [77]. These examples indicated that the expression of genes encoding three key heterologous enzymes has a significant role in ginsenoside production in engineered yeasts.

The introduction of a heterologous pathway is the most important part of metabolic engineering. However, the unbalanced cellular metabolic flux and the competitive consumption of intermediate and precursor metabolites are challenges for ginsenoside biosynthesis in engineered yeasts. Therefore, increasing isopentenyl diphosphate (IPP) and dimethylallyl diphosphate (DMAPP) in the mevalonate pathway (MVA) and improving the availability of 2,3-oxidosqualene from IPP/DMAPP provide the approaches to overcome the unbalanced cellular metabolic flux. Seven genes, including acetyl-CoA C-acetyltransferase (*ERG10*), HMG-CoA synthase (*ERG13*), mevalonate kinase (*ERG12*), phosphomevalonate kinase (*ERG8*), diphosphomevalonate decarboxylase (*ERG19*), isopentenyl diphosphate-isomerase (*IDI*), and 3-hydroxy-3-methylglutaryl-CoA reductase (*HMG*), belong to the MVA pathway for converting acetyl-CoA to IPP/DMAPP. Among the seven genes, HMG catalyzes the conversion of HMG-CoA to MVA, which plays a key role in the conversion of acetyl-CoA to IPP/DMAPP. It is demonstrated that the post-transcriptional feedback inhibition of HMG is the limited step in the MVA pathway. Therefore, one way to tackle this is to overexpress the truncated N-terminal of *HMG1* (*tHMG1*) along with the global transcription factor Upc2 of sterol biosynthesis. the lack of an N-terminal transmembrane sequence coding for the membrane-binding activity of tHMG1 combined with the sterol regulatory element-binding protein Upc2 resulted in improved ginsenoside production. For example, CK was obtained from *S. cerevisiae* harboring tHMG1 and Upc2 with 0.8 and 1.4 mg/L using glucose and galactose as carbon sources, respectively [70]. Similarly, PPD was achieved from *S. cerevisiae* harboring two transcriptional factors, *Upc2* and *INO2*, with 15.88 g/L using sugarcane molasses as a carbon source. the *INO2* gene, encoding for phospholipid biosynthesis, promotes and improves the catalytic efficiency of the endoplasmic reticulum (ER)-localized enzymes [78]. In another case, overexpressing genes in the MVA pathway, including *IDI1*, resulted in the improved production of 3β-O-Glc-DM and 20S-O-Glc-DM with 2400 and 5600 mg/L, respectively (Figure 3) [72]. In addition, overexpression of three genes, including farnesyl pyrophosphate synthetase (*ERG20*), farnesyl-diphosphate farnesyl transferase (also called squalene synthase) (*ERG9*), and squalene monooxygenase (also called squalene epoxidase) (*ERG1*), resulted in enhancing ginsenoside production through improving the conversion rate of the intermediate molecules such as FPP, squalene, and 2,3-oxidosqualene. As a result, the presence of *ERG20*, *ERG9*, and *ERG1* in engineered *S. cerevisiae* enhanced up to 10.9-fold the amount of DM with 10.97 mg/g DCW [79]. In general, the overexpression of related genes in the ginsenoside biosynthesis pathway is achieved through either integration into the yeast genome or plasmid systems.

**Table 3 molecules-28-01437-t003:** List of the engineered yeasts for ginsenoside production.

Strains	Genes or Related Gene Cassettes	Products	Titer (mg/L)	Cultivation Condition	Major Media	Carbon Source	References
** *Saccharomyces cerevisiae* **							
*ZD-PPD-018*	*tHMG1, AtCPR1, SynPgPPDS, ERG20, ERG1, ERG9*	*PPD*	*1189*	Fed-batch	SD	Glucose	[79]
		DM	1548				
D20RH18	*PgDDS, synPgPPDS, ATR2.1, tHMG1, ERG20, PgERG1, ERG9, UGTPg45*	Rh2	1.45 μmol/g DCW	Shake-flask	YPD	Glucose	[74]
D20RG1	*PgDDS, synPgPPDS, ATR2.1, tHMG1, ERG20, PgERG1, ERG9, UGTPg45, UGTPg29*	Rh3	3.49 μmol/g DCW				
ZW-Rh1-20	*ERG20, PgERG1, ERG9, tHMG1, CYP716A53v2, PgCPR1, UGTPg100*	Rh1	98.2	Shake-flask	SC	Glucose	[71]
		PPT	3.5				
		PPD	43.4				
		DM	8.8				
ZW-F1-17	*ERG20, PgERG1, ERG9, tHMG1, CYP716A53v2, PgCPR1, UGTPg1*	F1	42.1				
		PPT	13.9				
		CK	7.5				
		PPD	49.2				
		DM	3.5				
WLT-MVA5	*DS, PPDS-ATR1, ERG1, tHMG1, ERG9, ERG20, ERG10, ERG13, ERG12, ERG8, ERG19, IDI1, NCP1, ACS_seL641P_*	PPD	8090	Fed-batch	YNBD	Glucose/ Ethanol	[80]
Y1CSH	*HAC1, IDI1, ERG20, ERG9, ERG1, ERG7, synDS-GFP, tHMG1, synPgUGT74AE2*	3β-O-Glc-DM	2400	Fed-batch	YPD	Glucose	[72]
Y2CSH	*HAC1, IDI1, ERG20, ERG9, ERG1, ERG7, synDS-GFP, tHMG1, synUGTPg1*	20S-O-Glc-DM	5600				
PPD-A3-sgRNA4	*PgDS and PgPPDS, PgCPR, tHMGR1, ERG1m, ∆ ERG7*	PPD	294.5	Shake-flask	YPD	Glucose	[71]
Rg1-02	*CYP716A53v2, PgUGT71A54, PgURT94, RHM*	Rg2	1300	Fed-batch	Synthetic	Glucose	[75]
Re-01	*CYP716A53v2, PgUGT71A53, PgUGT71A54, PgURT94, RHM*	Re	3600				
CPX113436PPXP-ADH2	*ERG10, ERG13, tHMG1, ERG12, ERG8, ERG19, IDI1, ERG20, ERG9, ERG1, ERG7, PgDS, PgPPDS, PgCPR, ADH2, (Pex11p, Pex34p, and Atg36p)*	PPD	4.1	Shake-flask	YPDO	Glucose and Ethanol	[81]
BY-V	*ERG10, ERG13, tHMG1, ERG12, ERG8, IDI1, MVD1, ERG20, ERG9, ERG1, PgDDS, AtCPR1, PgPPDS, INO2, ∆LPP1, ∆ERG7*	PPD	1550	Shake-flask	YPD	Sugarcane molasses	[78]
			158,800	Fed-batch			
YFR	*tHMG1, IDI1, ERG20, ERG9, ERG1, DS-GFP, PGM1, PGM2, INO2, ERG7, ERG1, PgUGT74AE2, UGTPg1, PP-DS, ATR2*	F2	21.0	Shake-flask	YPD	Glucose	[73]
YSR	*tHMG1, IDI1, ERG20, ERG9, ERG1, DS-GFP, UGTPg1, PGM1, PGM2, INO2, ERG7, ERG1, M7 (ΔUGT74AC1)*	3β-20S-Di-O-Glc-DM	346.1			Glucose	
			2600	Fed-batch			
WEA	*tHMG1, ERG10, ERG13,IDI, ERG20, ERG9, ERG1, UGD1, AeBAS1, AtATR2, AeCYP716A354, AeCSLM1, AeUGT74AG6*	Chikusetsusaponin IVa	NR	Shake-flask	SD	Glucose, Galactose	[82]
	*tHMG1, ERG10, ERG13,IDI, ERG20, ERG9, ERG1, UGD1, AeBAS1, AtATR2, AeCYP716A354, AeCSLM1, AeUGT73CB3*	Zingibroside R1	NR				
*ZY-M7(4)E1* *PUA*	*ERG20, ERG1,* *ERG9, tHMG1,* *M7-1, ∆EGH1, PGM1,* *UGP1, PgPPDS-AtCPR2*	Rh2	300	Fed-batch	SC	Glucose	[83]
** *Yarrowia lipolytica* **							
Y14	*ΔLUL*, *XYL1*, *XYL2*, *ylXKS*, *DS*, *PPDS*-linker-*ATR1*, *tHMG1*, *ERG9*, *ERG20*, *TKL*, *TAL*, *TX*	PPD	300.63	Fed-batch	YPD or YPX	Xylose	[84]
			167.17			Glucose	
YL-MVA-CK	*tHMG1*, *ERG9*, *ERG20*, *OpDS*, *PPDS*-*linker2-ATR1*, *UGT1*	CK	161.8	Fed-batch	YPD	Glucose	[85]
** *Pichia pastoris* **							
KDPEP	*PgDDS*-L3-PDZlig and *ERG1*-ER/kPDZ with p-[*PgDDS*-PDZlig]/[*ERG1*-PDZ]	DM	0.10 mg/g DCW	Shake-flask	YPD	Glucose, methanol	[86]

SynPg: genes from Panax ginseng with codon optimization; Δ: mutant of genes; NR: not reported.

The recent demonstrations of engineering the ginsenoside pathway in microorganisms are promising; however, they are limited by the requirement of an expensive intermediate precursor exogenously. Acetyl-CoA is a key starting point of the MVA pathway for a variety of natural products. Therefore, the increasing acetyl-CoA level is a necessary prerequisite to enable the biosynthesis of ginsenoside from simple materials. Improving the supply of acetyl-CoA is achieved by overexpressing the endogenous gene encoding for the synthesis of acetyl-CoA. For example, a synthetic codon-optimized acetyl-CoA synthase mutant from *Salmonella enterica* (*ACSse_L641P_*) and NADP-dependent aldehyde dehydrogenase (*ALD6*) was expressed in *S. cerevisiae* WLT-MVA5, resulting in an increase of up to 66.55 mg/L of PPD production [80]. On the other hand, tricarboxylic acid (TCA) cycle, glycolysis pathway, glyoxylate cycle, and related amino acid metabolism are known as the competitive pathway with the MVA pathway in terms of acetyl-CoA consumption. Therefore, the deletion of genes involved in this pathway improved the accumulation of acetyl-CoA in the cytoplasm. For example, it is demonstrated that the production of DM in engineered *S. cerevisiae* YPH499 and INVSc1 was increased through the deletion of genes associated with amino acid metabolism, and mannose and fructose metabolism [87]. UDP-sugar is a key donor sugar in glycosylation reaction for ginsenoside production. Lack of a sugar donor such as UDP-glucose, UDP-Rhamnose, and UDP-glucuronic acid is one of the major limiting factors for the production of ginsenoside in yeasts. Therefore, increasing the UDP-sugar not only extends the various molecular ginsenosides, but also improves the productivity of ginsenoside in engineered yeasts. For example, overexpression of UDP-glucose 6-dehydrogenase 1 (*UGD1*) from *E. coli* and cellulose synthase-like M-subfamily genes from *Aralia elata* in engineered *S. cerevisiae* WEA resulted in producing more than 13 aralosides. In this case, UGD1 is catalyzed UDP-glucose to UDP-glucuronic acid, while *S. cerevisiae* WEA is an engineered strain for producing oleanolic acid [82]. Another example, overexpression of genes encoding for the UDP-glucose biosynthetic pathway from glucose *(HXK1* encoding hexokinase, *PGM1* encoding phosphoglucomutase, and *UGP1* encoding glucose-1 phosphate uridylyltransferase) in the engineered strain ZYM7 (4)-E –PU synthesized 65% more Rh2 than the parent strain with 36.7 mg/L (Table 3) [83]. Moreover, in the metabolic engineering of *S. cerevisiae*, including the reconstruction and expression of heterologous synthetic pathways, it is believed that the additional storage space of ginsenosides also resulted in improving the accumulation of target compounds. For example, modulating the expression of three peroxisomebiogenesis-related peroxins in *S. cerevisiae*, including peroxisome-population-regulated proteins (Pex11p and Pex34p) and autophagy-related protein (Atg36p), along with the expression of the PPD biosynthesis pathway in peroxisomes, enhanced more than a 78% higher level of PPD compared with the wild-type strain [81].

Recently, *Y. lipolytica* and *P. pastoris* have been considered realistic alternatives to *S. cerevisiae* for recombinant protein synthesis and natural bioactive production. In the case of ginsenoside production, CK could be obtained through overexpression of key genes in the MVA pathway and fusion of CYP450 and NADPH-P450 reductase in *Y. lipolytica*. As a result, the engineered *Y. lipolytica* YL-MVA-CK produced 161.8 mg/L of CK using glucose as a carbon source in fed-batch fermentation [85]. Like *S. cerevisiae*, both species show the ability to grow at high cell density with low nutritional supplies. However, *Y. lipolytica* and *P. pastoris* are highly metabolically adapted to catabolize feedstocks in bioprocesses. the engineered PPD-producing *Y. lipolytica* overexpressed endogenous xylulose kinase (*XKS*) and harbored xylose reductase (*XR*) and xylitol dehydrogenase (*XDH*), which produced 300.63 mg/L of PPD using xylose as a carbon source [84]. Similarly, a 2.1-fold of DM-II was enhanced in engineered *P. pastoris* through the expression and colocalization of two key enzymes, PgDDS and ERG1 (Table 3) [86].

## 5. Challenges and Future Perspectives

Microbial hosts have been shown to be the most advantageous for ginsenoside production. In the case of endophytes, endophytic bacteria and fungi have promise as novel sources of ginsenosides; however, only a few endophytes have shown the ability to synthesize ginsenosides. Therefore, the distribution, biodiversity, and composition of *ginseng* endophytes still must be investigated in depth to increase the number of potential endophytes from ginseng plants [88]. Recently, cultivation-independent approaches have been applied to the analysis of vast samples because they facilitate a rapid method. Furthermore, cultivation-independent methods allow a preliminary analysis of the associated community and draw a future strategy for traditional isolation techniques [89]. Noticeably, fluorescence in situ hybridization (FISH) and 16S rRNA gene metagenomics sequencing, are tools for the discovery of novel compounds from endophytic microorganisms in natural environments (Figure 4). Metagenomics techniques would provide a better understanding of the biodiversity and composition of endophytes, resulting in the potential discovery of new endophytes from *ginseng* plants [88,90]. Omics tools, including next-generation sequencing, metagenomics, comparative genomics, transcriptomics, proteomics, and metabolomics, have been applied to a deep understanding of plant-microbe interactions (Figure 4). It is believed that the application of the omics tool is a promising strategy to enhance ginsenoside production using microbial hosts [89,91]. the first metagenomics analysis of the bacterial endophyte community from *Pg* sheds light on endophyte–plant interaction. Moreover, the frequency of the putative functional gene showed the capability of endophytes for the solubilization of phosphate, methanol, and nitrogen metabolism. These might provide a new strategy to improve the production of ginsenosides from *Pg* endophytes [92]. Noticeably, transcriptomics and proteomic analysis of ginseng plants for ginsenoside biosynthesis have been strongly reported, but these data from ginseng endophytes are still lacking [93,94]. In general, ginsenosides are synthesized from two precursors, IPP and DMAPP, via the MEP and MVA pathways. Then, the biosynthesis of ginsenosides is catalyzed through three key enzymes, including oxidosqualene cyclases (OSCs), CYP450s, and GTs [95]. It is necessary to understand functional genes and enzymes from omics data. Because the absence of functional enzymes and the silencing of an acquired biosynthetic cluster in endophytes might be revealed. On the other hand, the β-glucosidase from endophyte could support the genes into recombinant strains, such as *E. coli* and/or *C. glutamicum.* This approach not only provides the high-efficiency production of ginsenoside, but also generates the various types of ginsenoside.

Since synthetic biology plays an important role in the engineering of model microorganisms, metabolic engineering-assisted synthetic biology provides a promising approach to producing ginsenosides in engineered microbes as well as endophytes. Synthetic biology tools can activate the silent gene clusters in endophytes. For example, a rate-limiting enzyme for the biosynthesis of taxol, namely taxadiene synthase, was expressed under the *trpC* promoter in the endophytic fungus EFY-21 (*Ozonium* sp.). As a result, the taxol production of endophytic fungus is 5-fold higher than that of wild-type endophytes [96]. Furthermore, functional and identified genes from *Panax* endophytes could be cloned and expressed in model microorganisms by using synthetic biology methods. This approach may be supported to generate unnatural ginsenosides [97]. Importantly, the development of synthetic biology tools, such as RNA interference (RNAi) and CRISPR-Cas systems, provides promising host strains with high titers of ginsenosides. CRISPR-Cas systems have emerged as a promising genome editing tool worldwide because of their not only simple design and high engineering feasibility, but also cost-effectiveness and high efficiency. Application of CRISPR-Cas technologies via knockdown, knockout, knockin, and fine-tuning of gene expression has been demonstrated to enhance the production of ginsenosides in *S. cerevisiae* [72,73]. Therefore, by employing the CRISPR-Cas system on yeasts, such as *Y. lipolytica*, *P. pastoris*, and endophytes, the high productivity of ginsenosides could be obtained.

Although bacterial endophytes produce more ginsenosides than fungal endophytes, the titer is still low. All fermentation experiments have been carried out under flask-shaking conditions and large-scale fermentation of endophytes. These approaches used a bioreactor, which has not been applied to the production of ginsenoside from endophytes yet. the maximum titers of Rg3 just peaked at 62.2 mg/L by using endophytic bacteria *Agrobacterium* sp., making this approach difficult for the production of industrial commodities [40]. One of the major reasons is the attenuation of ginsenoside production under artificial conditions without their host plants. the differences in environmental culture between host plant-associated endophytes and isolated endophytes from host plants reduce the potential for producing ginsenosides. the environmental factor includes carbon source, nitrogen source, pH, temperature, precursors, minerals, and trace elements. Therefore, the optimal process using bioreactors is expected to play an important role in increasing the production of ginsenosides from ginseng endophytes. In addition, the absence of selection pressure and coexisting endophytes might also be reasonable explanations for the loss of ginsenoside biosynthesis. A greater understanding of these aspects of endophytes would help to reduce attenuation and improve accumulation for ginsenoside biosynthesis. While endophytes provide a novel source of ginsenosides, engineered bacteria and yeasts provide approaches for the production of industrial commodities. Although 15.8 g/L of PPD and 8.088 g/L of DM were achieved by combining all engineered strategies, the large capacity of ginsenoside production is still a high requirement [79,98]. the optimal process of the fermentation system is also required for engineered strains, especially its carbon and energy source. Unlike *S. cerevisiae*, the growth ability in a wide pH range of *Y. lipolytica* and *P. pastoris* helps both strains being more favorable hosts for ginsenoside production. Moreover, the ability to use methanol as a carbon and energy source led to *P. pastoris* becoming one of the most important alternative strains for ginsenoside production.

## 6. Conclusions

Ginsenosides have a great and significant effect on human health. Whole-cell endophytic microbes isolated from the family *Araliaceae* have recently been developed for the production of ginsenosides. the high-value ginsenosides could be obtained through either native endophytes or biotransformation from major ginsenosides. To date, there is no industrial process for the production of ginsenosides using ginseng endophytes. However, endophytes not only provide a novel platform for ginsenoside biosynthesis, but also open up an eco-friendly source of natural compounds. the interaction between endophytes and host plants must be investigated at the molecular level. These strategies may help create optimal conditions for fermentation. Moreover, a greater understanding of endophyte genomics would help to express the silence gene cluster for ginsenoside production. Importantly, the application of omics tools and synthetic biology are expected to be novel approaches for overcoming the disadvantages of using microbial endophytes in the production of ginsenosides.

Metabolic engineering of microorganisms has been recognized as a sustainable approach to producing natural and unnatural ginsenosides. Although whole-genome sequences of model microbes have been identified, *E. coli* and *C. glutamicium* are the most commonly used for the expression of β-glucosidase recombinant. On the other hand, *S. cerevisiae* and nonconventional yeasts (*Y. lipolytica* and *P. pastoris*) have been carried out as ginsenoside cell factories. Various metabolic engineering methods have been developed and applied to achieve the efficient production of ginsenosides in yeasts. These approaches include heterologous gene expression, enzyme engineering, balancing and increasing metabolic flux, and fermentation strategies. Interestingly, metabolic engineering-assisted omics and metabolic engineering assisted by synthetic biology can accelerate the enhancement of yeast cell factories for industrial applications.

## Figures and Tables

**Figure 1 molecules-28-01437-f001:**
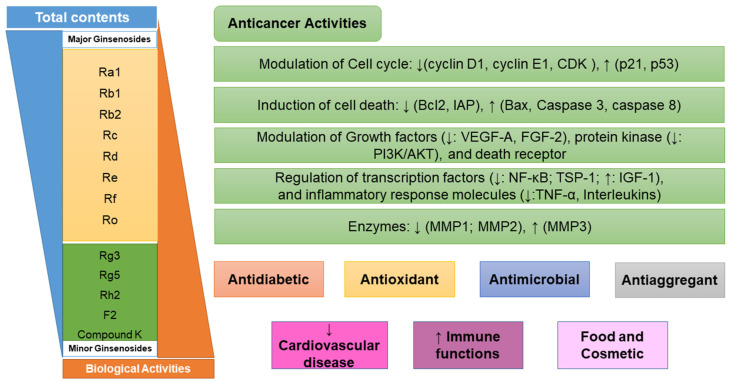
Biological activities and cellular mechanisms versus cancer of ginsenosides. CDK: cyclin-dependent kinase; IAP: inhibitory apoptotic protein; VEGF-A: vascular endothelial growth factor A; FGF-2: fibroblast growth factor 2; PI3K/AKT: protein kinase B signaling pathway; TSP-1: inhibitors thrombospondin-1; IGF-1: insulin-like growth factor-1; TNF-*α*: tumor necrosis factor-α; MMP: matrix metalloproteinase; NF-*κ*B, nuclear factor *κ*B.

**Figure 2 molecules-28-01437-f002:**
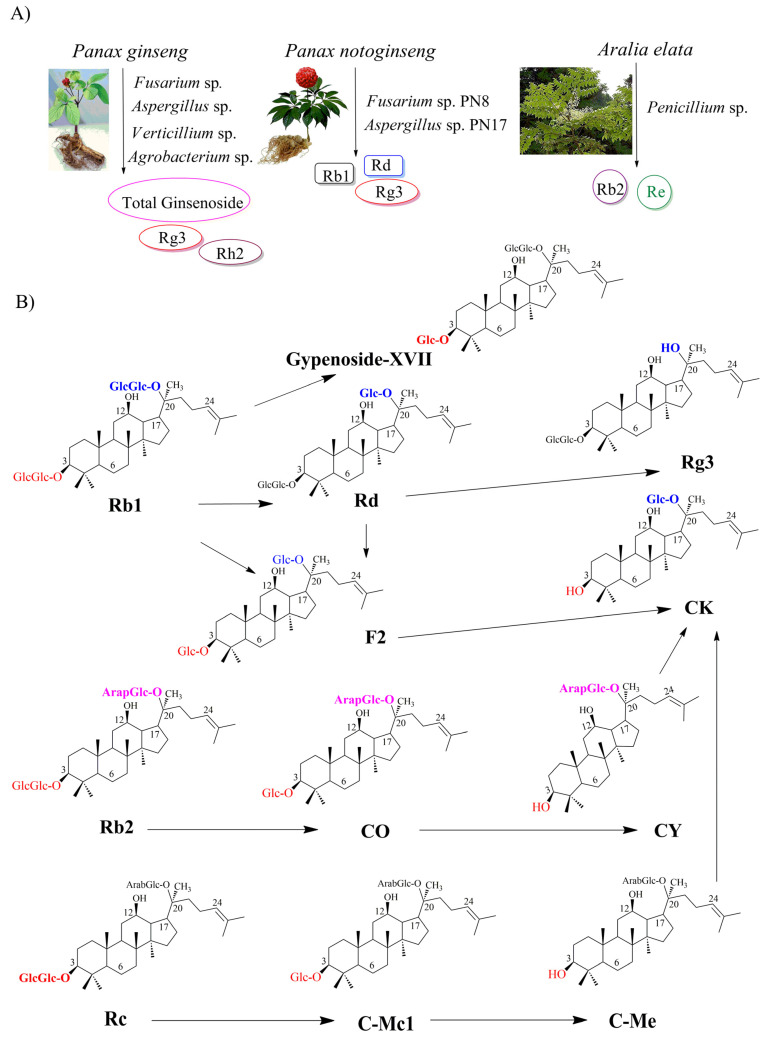
Bioproduction of ginsenosides by endophytes. (**A**) Native endophytes as biological source of ginsenosides; (**B**) Biosynthesis of ginsenosides from major ginsenosides through biotransformation.

**Figure 3 molecules-28-01437-f003:**
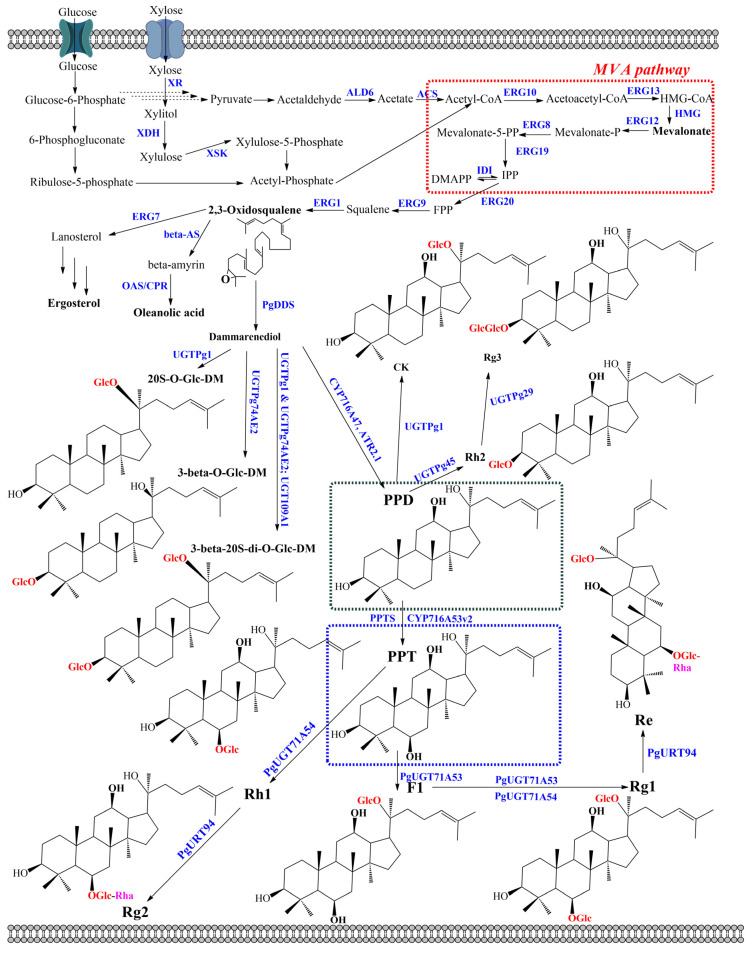
Biosynthetic pathway for ginsenoside production in cytosol of engineered yeasts. Enzymes: ACS, acetyl-CoA synthase; ALD6, acetaldehyde dehydrogenases encoding by *ald6*; beta-AS, β-amyrin synthase; CPR, cytochrome P450 reductase; CS, cycloartenol synthase; DDS, dammarenediol-II synthase; ERG1, squalene epoxidase; ERG7, lanosterol synthase; ERG8, phosphomevalonate kinase; ERG9, squalene synthase; ERG10, acetyl-CoA C-acetyltransferase; ERG12, mevalonate kinase; ERG13, HMG-CoA synthase; ERG19, diphosphomevalonate; ERG20, farnesyl-diphosphate synthase; IDI, isopentenyl diphosphate-isomerase; HMG, 3-hydroxy-3-methylglutaryl-CoA reductase; OAS, oleanolic acid synthase; PPTS, protopanaxatriol synthase; XDH, xylitol dehydrogenase; XKS, Xylulose kinase; XR, xylose reductase.

**Figure 4 molecules-28-01437-f004:**
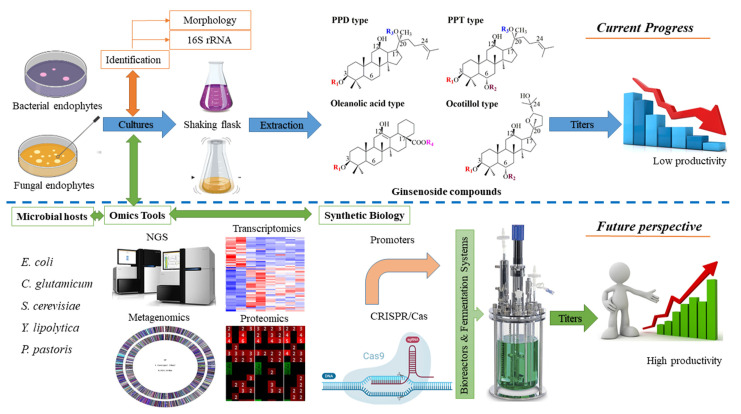
Processing and future perspective of production ginsenosides by microbial hosts.

**Table 1 molecules-28-01437-t001:** Classification and cell biological mechanism in anticancer activities of four types of ginsenosides.

Structure	Name	R1	R2	R3	R4	Cellular Mechanisms	Ref.
**Protopanaxadiol (PPD) Type**							
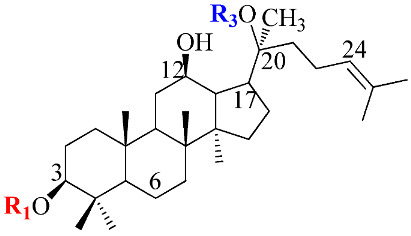	Ra1	Glc^2^-Glc	-	Glc^6^-Ara(p)^4^-Xyl	-	Not reported	
	Rb1	Glc^2^-Glc	-	Glc^6^-Glc	-	Inhibition of invasion and migration	[17,21]
	Rb2	Glc^2^-Glc	-	Glc^6^-Ara(p)	-	Inhibition of metastasis and proliferation	[17]
	Rc	Glc^2^-Glc	-	Glc^6^-Ara(f)	-	Anti-proliferative activity	[19]
	Rd	Glc^2^-Glc	-	Glc	-	Inhibit proliferation; Inhibit angiogenesis	[4]
	Rg3	Glc^2^-Glc	-	H	-	Repression of cell proliferation and induce apoptosis	[20,25]
	Rh2	Glc	-	H	-	Modulation of cell cycle; Regulation of inflammatory response molecules	[20,27]
	F2	Glc	-	Glc	-	Inhibit proliferation	[4]
	CK	H	-	Glc	-	Modulation of growth factors and regulation of transcription factors; Induce apoptosis	[5,23]
**Protopanaxatriol (PPT)-type**							
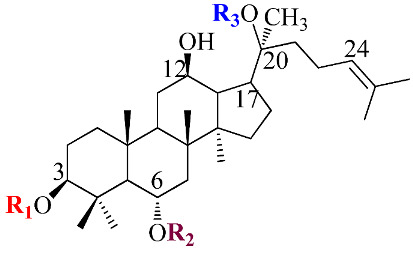	Re	OH	Glc^2^-Rha	Glc	-	Not reported	
	Rf	OH	Glc^2^-Glc	H	-	Cell cycle arrest and apoptosis	[19]
	Rg1	OH	Glc	Glc	-	Induce apoptosis; Repression of cell proliferation	[4,21]
	Rg2	OH	Glc^2^-Rha	H	-	Induce apoptosis; Repression of cell proliferation	[4]
	Rh1	OH	Glc	H	-	Regulation of gene coding for metalloproteinase; Repression of cell proliferation	[5,28]
	F1	OH	H	Glc	-	Modulation of death receptor	[29]
	Notoginsenoside R1	H	Glc^2^-Xyl Glc	Glc	-	Regulation of inflammatory response molecules	[19]
**Ocotillol-type**							
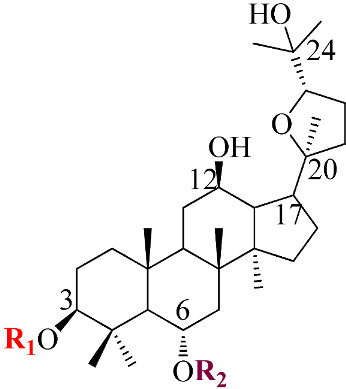	Majonoside R2	OH	Glc^2^-Xyl	-	-	Not reported	
	Vinaginsenoside R1	OH	Ac-Glc^2^-Rha	-	-	Not reported	
**Oleanolic acid type**							
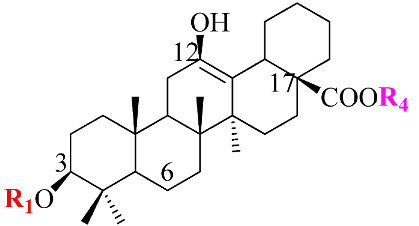	R_O_	GlcUA-Glc	-	-	Glc	Not reported	
	R_OA_	GlcUA-Glc	-	-	Glc^6^-Glc	Not reported	

Ac: acetyl; Ara(p): α-L-glucopyranosyl; Ara(f): α-L-arabinofuranosyl; Glc: β-D-glucopyranosyl; Rha: α-L-rhamncpyranosyl.

**Table 2 molecules-28-01437-t002:** Summary of ginsenoside production by endophytes in Shake-flask cultures.

**Host**	**Endophytic Strains**	**Type of Compounds/** **Biotransformation** **Pathway**	**Major Media**	**Titer** **(mg/mL)**	**References**
** *Ginsenoside production by native endoyphtes* **					
*Aralia elata*	*Penicillium* sp.	Rb2, Re	PDA liquid	2.049	[41]
*P. ginseng*	*Fusarium* sp.	Total ginsenoside	PDA liquid	0.181	[40]
	*Aspergillus* sp.			0.144	
	*Verticillium* sp.			0.144	
*P. notoginseng*	*Fusarium* sp. PN8	Rb1, Rd, and Rg3	PDA liquid	1.061	[39]
	*Aspergillus* sp. PN17	Re, Rd, and Rg3		0.583	
*P. ginseng*	*Agrobacterium* sp.	Rg3	LL medium	62.20 mg L^−1^	[42]
		Rh2		18.60 mg L^−1^	
** *Biotransformation of major to rare ginsenosides by endophytes* **					
*P. ginseng*	*Arthrinium* sp.	Rb1 → Rd → F2 → CK	PDA liquid	NA	[47]
	*Burkholderia* sp.	Rb1 → Rd → Rg3	PDA liquid	NA	[48]
	*Flavobacterium* sp.	Rb1 → Gyp-XVII	PDA liquid	NA	[49]
*Platycodon grandiflorum*	*Luteibacter* sp.	Rb1 → Rd → F2	LL medium	0.06692	[50]
		Rb1 → Rd → F2 → CK		0.03323	
		Rb2 → CO → CY → CK			
		Rc →CMc1 → CMe → CK			
		Rg1 → Rh1		NA	
*P. notoginseng*	*Fusarium oxysporum* or *Fusarium* sp.	Rb1 → CK	LB medium	0.02	[38]
		Rb1 → F2		0.025	
	*Nodulisporium* sp.	Re → 6-O-[*α*-L-Rhamnopyranosyl-(1→2)-*β*-D-glucopyranosyl]-20-O-*β*-glucopyranosyl-dammarane-3,6,12,20,24,25-hexaol		0.125	
		Vinaginsenoside R13		0.09	
	*Fusarium oxysporum Nodulisporium* sp. *Bacillus* sp.	Rg1 → Vinaginsenoside R22		0.065	
	*Nodulisporium* sp.	Rh1 → Pseudo-ginsenoside RT4		0.075	
	*Fusarium oxysporum*	Rh1 → PPT		0.02	
	*Brevundimonas* sp.	Rh1 → Rg1		0.15	
		Rh1 → Vinaginsenoside R15		0.05	
	*Bacillus* sp.	Rh1 → (20S)-3-O-*β*-D-glucopyranosyl-6-O-*β*-D-glucopyranosylprotopanaxatriol		0.07	
*P. notoginseng*	*Enterobacter* *chengduensis*	Rg1 → F1	PDA medium	13.24%;	[46]
	*Trichoderma koningii*	Rb1 → Rd		40.00%	
		Rb1 → Rg3		32.31%;	
	*Penicillium chermesinum*	Rb1 → Rd		74.24%	
*P. quinquefolius*	*Bacillus* sp. G9y	Rc → Rd	Beef extract peptone	100%	[51]

## Data Availability

The datasets/materials generated and analyzed during the current study are available on request from the corresponding author.

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
