# Peer review of "Microorganisms for Ginsenosides Biosynthesis: Recent Progress, Challenges, and Perspectives"

_molecules, 2023, doi:10.3390/molecules28031437_

Round 1

Reviewer 1 Report

Microorganisms for Ginsenosides Biosynthesis: Recent Progress,

Challenges, and Perspectives

Submitted to Molecules (molecules-2153876)

The review article summarized cell biological mechanism in anticancer activities and the present state of research on the production of ginsenosides. Interestingly, native endophytes have been used as a novel source for ginsenoside biosynthesis. Furthermore, it is expected that engineered microbial hosts (Escherichia coli, Corynebacterium glutamicum, Saccharomyces cerevisiae, Yarrowia lipolytica, and Pichia pastoris) provided a promising strategy for producing a high quantity of ginsenosides.

Overall, the topic the authors reviewed is very important for researchers in microbial products, metabolic engineering and synthetic biology. Moreover, the structure is well-organized. The current version needs a few revisions and it should be accepted for publication.

The listed are some comments regarding the submitted manuscript:

1.      All Panax genus should be written on an abbreviation (For example, Line 304, page 8 – Panax bipinnatifidus var. bipinnatifidus, → P. bipinnatifidus var. bipinnatifidus; Line 307- page 8: Panax ginsengP. ginseng).

2.      Line 312: Since the E. coli is known as the most important microbial host for industrial production, please provide the advantages of E. coli for producing ginsenoside.

3.      Line 655: Oikos → Oikos

4.      Line 752: Molecules, 2017 → Molecules 2017

5.      Line 883: Commun Biol.Commun. Biol.

This paper should be accepted for publication after minor revision.

Author Response

Dear Reviewer

Thank you so much for reviewing our manuscript "Microorganisms for Ginsenosides Biosynthesis: Recent Progress, Challenges, and Perspectives" (molecules-2153876) and finding it potentially suitable for publication in Molecules.  We really appreciate all your generous comments and suggestions! Those comments and suggestions are all valuable and very helpful for revising and improving our paper. 

We have addressed most of the reviewers’ comments. The point-by-point responses to the comments are also given in red fonts and changes made on the manuscript are also marked with blue font for easy review process. The manuscript has been edited by a professional English editor prior to revise submission.

Response to Reviewer Comments

  1. All Panaxgenus should be written on an abbreviation (For example, Line 304, page 8 – Panax bipinnatifidus bipinnatifidus, → P. bipinnatifidus var. bipinnatifidus; Line 307- page 8: Panax ginseng → P. ginseng).

Response: We have corrected in the revised manuscript.

  1. Line 312: Since the  coliis known as the most important microbial host for industrial production, please provide the advantages of E. coli for producing ginsenoside.

Response: We have provided the advantages of E. coli for producing ginsenoside in the text.

"Although E. coli lacks endoplasmic reticulum required for plant membrane-bound cytochrome P450 enzymes (CYP450s) and is less efficient in precursor supplies for the methylerythritol phosphate (MEP) pathway, E. coli is known as the most important microbial host for industrial production. E. coli have many dominant characteristics compared to plant cells, such as fast growth with high cell density cultivation, controllability under laboratory conditions, well-characterized genetics, and well-developed genetic manipulation technology."

  1. Line 655: Oikos → Oikos

Response: We have corrected in the revised manuscript.

  1. Line 752: Molecules, 2017 → Molecules 2017

Response: We have corrected in the revised manuscript.

  1. Line 883: Commun Biol. →  Commun. Biol.

 Response: We have corrected in the revised manuscript.

Reviewer 2 Report

The importance of ginsenosides as drugs or nutraceuticals cannot be overstated, and traditional plant sources have many drawbacks. The use of microorganisms to manufacture ginsenosides has several advantages, not only in alleviating source scarcity, but also in extracting and isolating them in an efficient and time-saving manner. This manuscript reviews the current status of research on the use of microorganisms to manufacture ginsenosides and discusses the current challenges and prospects of using microbial hosts for ginsenoside production.

The manuscript is well written, logical and well organized. Figure 4 summarizes well.

This manuscript could be considered for publication in this journal.

Including but not limited to the following details issues still need to be fixed.

1. Figure 2B, the logic is confusing and the information conveyed is not well expressed.

2. the literature collection is not comprehensive, such as this article "Deletion and tandem duplications of biosynthetic genes drive the diversity of triterpenoids in Aralia elata ".

Author Response

Dear Reviewer

Thank you so much for reviewing our manuscript "Microorganisms for Ginsenosides Biosynthesis: Recent Progress, Challenges, and Perspectives" (molecules-2153876) and finding it potentially suitable for publication in Molecules. We really appreciate all your generous comments and suggestions! Those comments and suggestions are all valuable and very helpful for revising and improving our paper.

We have addressed most of the reviewers’ comments. The point-by-point responses to the comments are also given in red fonts and changes made on the manuscript are also marked with blue font for easy review process. The manuscript has been edited by a professional English editor prior to revise submission.

Response to Reviewer Comments

  1. Figure 2B, the logic is confusing and the information conveyed is not well expressed.

Response: Thank you for your suggestion and we apologize for our carelessness. We have re-draw and added the legend of Figure 2 to provide accurate information.

  1. the literature collection is not comprehensive, such as this article "Deletion and tandem duplications of biosynthetic genes drive the diversity of triterpenoids in Aralia elata ".

Response: Thank you for careful reading of our manuscript and for your comment for in your busy schedule. We refer to other literature for the construction of the idea of the article [83,84].

“UDP-sugar is a key donor sugar in glycosylation reaction for ginsenoside production. Lack of sugar donor as UDP-glucose, UDP-Rhamnose, and UDP-glucuronic acid is one of the major limiting factors for production of ginsenoside in yeasts. Therefore, increasing the UDP-sugar not only extends the various molecular ginsenosides but also improves the productivity of ginsenoside in engineered yeasts. For example, overexpression of UDP-glucose 6-dehydrogenase 1 (UGD1) from E. coli and cellulose synthase-like M-subfamily genes from Aralia elata in engineered S. cerevisiae WEA resulted in producing more than 13 aralosides. In this case, UGD1 is catalyzed UDP-glucose to UDP-glucuronic acid while S. cerevisiae WEA is engineered strain for producing oleanolic acid [83]. Another example, overexpression of genes encoding for UDP-glucose biosynthetic pathway from glucose (HXK1 encoding hexokinase, PGM1 encoding phosphoglucomutase, and UGP1 encoding glucose-1 phosphate uridylyltransferase) in the engineered strain ZYM7 (4)-E –PU synthesized 65% more Rh2 than the parent strain with 36.7 mg/L [84].

References

 [83] Wang, Y.; Zhang, H.; Ri, H. C.; An, Z.; Wang, X.; Zhou, J. N.; Zheng, D.; Wu, H.; Wang, P.; Yang, J.; Liu, D. K.; Zhang, D.; Tsai, W. C.; Xue, Z.; Xu, Z.; Zhang, P.; Liu, Z. J.; Shen, H.; Li, Y. Deletion and tandem duplications of biosynthetic genes drive the diversity of triterpenoids in Aralia elata. Nat. Commun. 2022, 13(1), 2224.

[84] Zhuang, Y.; Yang, G. Y.; Chen, X.; Liu, Q.; Zhang, X.; Deng, Z.; Feng, Y. Biosynthesis of plant-derived ginsenoside Rh2 in yeast via repurposing a key promiscuous microbial enzyme. Metab. Eng. 2017, 42, 25-32.

Reviewer 3 Report

Title:  Microorganisms for Ginsenosides Biosynthesis: Recent Progress, Challenges, and Perspectives

The review summarizes the different biological properties and cellular mechanisms of ginsenosides as natural anticancer agents. Furthermore, this work aims to describe the state of the art of ginsenoside biosynthesis and production by native endophytes, a new platform for metabolite production, and engineered microorganisms, that provide an high efficiency production.

The subject of the review is interesting and the description of the state of the art is good; unfortunately the whole work is poorly written and needs a very deep revision of the English and the structure of the paper is not well organized, the chapters lack subsections that could make the work easier to read. As it stands it is incomprehensible.

Points to review:

·        Introduction (Lines 50-76) the sentences are not grammatically correct, the verbs are missing and the sentences are often meaningless and difficult to understand. please rephrase all introduction.

·        Lines 84-91: it is advisable to refer to a figure (new or already present) which shows the chemical structures to make it easier to understand.

·        Lines 140-163 all this information would be better summarized in a table to make it more understandable and with a better visual impact.

·        Figure 2B: the graphical representation of the pathway is unclear. Too confused.

·        In table 1, the column referred to cultivation condition does not provide any additional information; it was enough to write in the title of the table that all experiments were conducted in shake-flask!

In addition, in my opinion, there is also a missing part referring to the extraction of ginsenosides from engineered cells; from figure 3, it appears that the ginsenosides are not excreted from the cells

Author Response

Dear Reviewer

Thank you so much for reviewing our manuscript "Microorganisms for Ginsenosides Biosynthesis: Recent Progress, Challenges, and Perspectives" (molecules-2153876) and finding it potentially suitable for publication in Molecules. We really appreciate all your generous comments and suggestions! Those comments and suggestions are all valuable and very helpful for revising and improving our paper.

We have addressed most of the reviewers’ comments. The point-by-point responses to the comments are also given in red fonts and changes made on the manuscript are also marked with blue font for easy review process. The manuscript has been edited by a professional English editor prior to revise submission.

Response to Reviewer Comments

  • Introduction (Lines 50-76) the sentences are not grammatically correct, the verbs are missing and the sentences are often meaningless and difficult to understand. please rephrase all introduction.

Response: Thank you for your suggestion and we apologize for our carelessness. We have corrected all introductions.

  • Lines 84-91: it is advisable to refer to a figure (new or already present) which shows the chemical structures to make it easier to understand.

Response: We have provided Table 1 which showed the chemical structures of four group’s ginsenosides.

  • Lines 140-163 all this information would be better summarized in a table to make it more understandable and with a better visual impact.

Response: We have summarized the possible cell biological mechanism of action of ginsenosides on Table 1 and Figure 1.

  • Figure 2B: the graphical representation of the pathway is unclear. Too confused.

Response: Thank you for your suggestion and we apologize for our carelessness. We have re-draw and added the legend of Figure 2 to provide accurate information.

  • In table 1, the column referred to cultivation condition does not provide any additional information; it was enough to write in the title of the table that all experiments were conducted in shake-flask!

Response: We have written the cultivation condition in the title of the Table 2.

In addition, in my opinion, there is also a missing part referring to the extraction of ginsenosides from engineered cells; from figure 3, it appears that the ginsenosides are not excreted from the cells

Response: We focused to describe the biosynthesis pathway for ginsenoside production in cytoplasmic matrix of yeasts. Therefore, we have not referred to the extraction of ginsenosides from engineered cells in Figure 3. According to your opinion, we have re-written the legend of Figure 3 to provide accurate information.

Reviewer 4 Report

The manuscript is well prepared. The authors carefully selected the literature. however, the authors should add figures with the chemical structures of the compounds. Giving names in the text reduces its readability.

Author Response

Dear Reviewer

Thank you so much for reviewing our manuscript "Microorganisms for Ginsenosides Biosynthesis: Recent Progress, Challenges, and Perspectives" (molecules-2153876) and finding it potentially suitable for publication in Molecules. We really appreciate all your generous comments and suggestions! Those comments and suggestions are all valuable and very helpful for revising and improving our paper.

We have addressed most of the reviewers’ comments. The point-by-point responses to the comments are also given in red fonts and changes made on the manuscript are also marked with blue font for easy review process. The manuscript has been edited by a professional English editor prior to revise submission.

Response to Reviewer Comments

The manuscript is well prepared. The authors carefully selected the literature. however, the authors should add figures with the chemical structures of the compounds. Giving names in the text reduces its readability.

Response: We have provided Table 1 which showed the chemical structures of four group’s ginsenosides.

Round 2

Reviewer 3 Report

Thanks to the authors for having fully answered my questions and for having carefully corrected the paper, improving it both graphically and in English. Particularly:

- the introduction primarily and all the text have been rewritten correctly, now making the paper easy to read and understand

- table 1, which has been added, helps the reader a lot in understanding the subject matter.

Minor errors to correct:

Line 286 correct “delycosylation

Line 435 “ò” change to ”of”

Line 580 correct “oxiosqualene”

Line 610 correct “CRISIR-Cas”

Line 610 yeast are not “bacterial microbes”

Author Response

Dear Reviewer

Thank you so much for reviewing our manuscript "Microorganisms for Ginsenosides Biosynthesis: Recent Progress, Challenges, and Perspectives" (molecules-2153876) and finding it potentially suitable for publication in Molecules. We really appreciate all your generous comments and suggestions! Those comments and suggestions are all valuable and very helpful for revising and improving our paper.

We have addressed most of the reviewers’ comments. The point-by-point responses to the comments are also given in red fonts and changes made on the manuscript are also marked with blue font for easy review process. The manuscript has been edited by a professional English editor prior to revise submission.

Response to Reviewer Comments

 Line 286 correct “delycosylation

Response:  Line 286: delycosylation → deglycosylation

Line 435 “ò” change to ”of”

Response:  Line 435: “ò” →”of”

Line 580 correct “oxiosqualene”

Response:  Line 580: oxiosqualene → oxidosqualene

Line 610 correct “CRISIR-Cas”

Response:  Line 610: CRISIR-Cas → CRISPR-Cas

Line 610 yeast are not “bacterial microbes”

Response:  Line 610: bacterial microbes → yeasts